# Viscoelastic Properties of Human Facial Skin and Comparisons with Facial Prosthetic Elastomers

**DOI:** 10.3390/ma16052023

**Published:** 2023-02-28

**Authors:** Mark W. Beatty, Alvin G. Wee, David B. Marx, Lauren Ridgway, Bobby Simetich, Thiago Carvalho De Sousa, Kevin Vakilzadian, Joel Schulte

**Affiliations:** 1Research Service, VA Nebraska-Western Iowa Healthcare System, 4101 Woolworth Avenue, Omaha, NE 68105, USA; 2Department of Adult Restorative Dentistry, University of Nebraska Medical Center College of Dentistry, 4000 East Campus Loop South, Lincoln, NE 68583, USA; 3Department of Restorative Sciences, University of Minnesota School of Dentistry, Malcolm Moos Health Sciences Tower, 515 Delaware Street SE, Minneapolis, MN 55455, USA; 4Department of Statistics, 340 Hardin Hall, University of Nebraska-Lincoln, Lincoln, NE 68583, USA; 5Formerly Department of Prosthodontics, Creighton University School of Dentistry, 2109 Cuming Street, Omaha, NE 68102, USA; 6Department of Dentistry, School of Health Sciences, University of Brasilia (UnB), Brasilia 70910-900, Brazil; 7Private Practice, Pine Ridge Dental, 8545 Executive Woods Drive Suite #2, Lincoln, NE 68512, USA; 8Process Engineer, GSK Consumer Healthcare, 1401 Cornhusker Highway, Lincoln, NE 68517, USA

**Keywords:** maxillofacial prosthesis, skin aging, face, viscoelastic substances, elasticity, siloxanes

## Abstract

Prosthesis discomfort and a lack of skin-like quality is a source of patient dissatisfaction with facial prostheses. To engineer skin-like replacements, knowledge of the differences between facial skin properties and those for prosthetic materials is essential. This project measured six viscoelastic properties (percent laxity, stiffness, elastic deformation, creep, absorbed energy, and percent elasticity) at six facial locations with a suction device in a human adult population equally stratified for age, sex, and race. The same properties were measured for eight facial prosthetic elastomers currently available for clinical usage. The results showed that the prosthetic materials were 1.8 to 6.4 times higher in stiffness, 2 to 4 times lower in absorbed energy, and 2.75 to 9 times lower in viscous creep than facial skin (*p* < 0.001). Clustering analyses determined that facial skin properties fell into three groups—those associated with body of ear, cheek, and remaining locations. This provides baseline information for designing future replacements for missing facial tissues.

## 1. Introduction

Skin consists of an outer layer of stratified keratinized epithelium; a middle dermis layer consisting of fibrous, collagenous, and elastic tissue; and a deep hypodermis layer that primarily contains pads of adipose tissue. Skin with a thick epidermal layer (0.8–1.4 mm) is found on the palms of hands and soles of feet, whereas thin epidermis (0.07–0.12 mm) is present in other locations, including the face. Thin epidermis is comprised of four strata (corneum, granulosum, spinosum, and basale). Dermis has an upper papillary layer with connective tissue cells, collagen type III, and a loose elastic fiber network. It is supported by a lower dense reticular layer consisting of collagen type I fiber bundles and coarse elastic fibers. The deep hypodermis is made up of loose connective tissue that changes into adipose tissue of varying thickness depending upon body location and sex. It stores energy and absorbs shock. Facial skin contains sebaceous glands, which provide an oily surface. It also receives the most sun exposure, and its properties are related to the extent of actinic damage. A more detailed description of facial skin histology and function can be found in the publication by Arda et al. [1]. Depending upon the measurement method used, facial skin thickness has been reported to be between 0.4 mm and 2.5 mm, and varies by location [2,3,4,5,6].

The biomechanical properties of skin are attributed primarily to the connective tissue present in the dermis. The physical organization and relative amounts of collagen, elastin, glycosaminoglycans, and water contribute to the non-linear stress–strain properties that are accompanied by hysteresis. Time-dependent deformation of skin is characterized by elastic deformation that is followed by viscoelastic creep, and an immediate elastic recovery followed by creeping recovery with residual deformation [7]. Due to the collagenous network oriented along Langer’s lines, skin is anisotropic and behaves differently along different loading directions.

Facial disfigurement arises from trauma, burns, and surgical removal of tumors. It is estimated that annually 400,000 civilian cases of facial fractures are treated in the United States, one to five million burns to the head and neck occur worldwide, and 850,000 new cases of head and neck cancer are diagnosed across the globe [8,9,10]. Treatment modalities include reconstructive surgery, placement of tissue-engineered constructs and biomimetics, and construction of maxillofacial prostheses. For cases where facial disfigurement results, a disfigured person is perceived as less attractive, less intelligent, less capable, and unemployable [11]. This contributes to a lowered self-esteem, which negatively affects the quality of life [12].

Reconstructive surgical techniques have limitations due to their reliance on autogenous and allogeneic materials. When used to replace hard and soft tissues, they are in short supply, may not conform to the intricate geometry required to replicate missing tissue, undergo wound contracture, and, if used as grafts, contribute to donor site morbidity [13]. For this reason, research has focused on replacing missing facial tissues through a combinatorial approach that includes tissue engineering, biomimetics, and prosthetics. Since a desired endpoint is the duplication of the appearance, biology, and function of native tissue, successful engineering of a biological or synthetic replacement requires a set of well-defined parameters known for healthy tissue.

For patients who wear facial prostheses, improvements in pigmentation systems have rendered them aesthetically acceptable. Facial skin tones that are customized chairside produce a prosthesis that has a natural look at the time of delivery. However, polymers used for constructing these prostheses are based on filled poly(dimethyl siloxanes) or polyurethanes that render prostheses having high tensile stiffness and low damping properties. As a result, patient dissatisfaction with comfort and wearability has been reported, along with longevity, function, and discoloration. It has been documented that up to 12% of patients will not wear their prostheses [14,15].

Reports of facial skin properties are limited, and comparisons with mechanical properties of prosthetic elastomers are virtually non-existent. Published data are limited by measurements of cadaveric tissue [16], small numbers of human subjects [17,18], and selection of relatively few facial locations [5,19,20]. What is needed are data derived from vital facial skin at locations where injury occurs. The purpose of this study was to measure viscoelastic properties in facial skin in a subject population equally stratified by age, gender, and race, at six different facial locations. Then, the results were compared with those measured for a group of prosthetic elastomers currently available for clinical usage. The null hypothesis tested was that there are no differences in biomechanical properties in facial skin based on facial location, sex, age, or race. A second hypothesis was that there are no differences between facial skin properties and those measured for prosthetic elastomers.

## 2. Materials and Methods

### 2.1. Facial Skin Measurements

A total of 144 human subjects were enrolled and stratified to produce equal numbers of subjects represented by gender (male, female), age (19–29, 30–49, and 50–70) and race/ethnicity (Asian, African American, Hispanic, and White) (*n* = 6). The sample size was calculated from power analyses of preliminary trial data to detect a significant difference (*p* < 0.05) of 3 kPa/mm stiffness at different facial locations with 80% statistical power. The study enlisted 52 United States military veterans and 92 non-veterans as participants. All protocols for patient recruitment, informed consent, privacy, and data security were approved by the Omaha VA Subcommittee for Human Studies (IRB Project #00644, approved 11 August 2011), and the Creighton University Institutional Review Board (#13-16724, approved 13 June 2013). Inclusion criteria included healthy facial skin that was not covered with scars or skin lesions where skin measurements were to be made. Exclusion criteria included scarring, facial hair, or prosthesis present in measurement areas. Subjects were asked to refrain from applying facial cream 24 h in advance of the study, and to remove facial jewelry during measurement. After obtaining written informed consent, skin measurements were made at six locations: cheek, chin, tip of the nose, forehead, ear lobe, and body of the ear. An alcohol wipe was used to cleanse the measurement area, and an erasable marker was used to mark locations for the chin, nose, and forehead at approximately the facial midline. Cheek measurements were made two centimeters lateral to the left ala of the nose. Ear lobe measurements were made on the anterior surface of the left ear lobe. Body of left ear measurements were made on the flattest region on the back side, adjacent to the helix and beneath the antihelical fold. Biomechanical measurements were made by placing a hand-held glass chamber with a 1 cm diameter port on the face (BTC-2000, SRLI Technologies, Figure 1), and a vacuum was applied to the skin at a rate of 1.33 kPa/s over a 15 s time period until a maximum (negative) pressure of 20 kPa was reached. The vacuum was held an additional 10 s, the vacuum released, and skin deformation (creep) measured for an additional three seconds.

Stress-displacement and deformation-time data were recorded, and six biomechanical properties determined: percent laxity, stiffness, elastic deformation, creep, absorbed energy, and percent elasticity (Figure 2, following page).

Stress was recorded in units of mm Hg and converted to kPa by multiplying each mm Hg by 0.1333 to achieve the number of kilopascals. No skin thickness measurements were taken. Descriptive statistics and a four-factor linear mixed model with full interaction were applied to each biomechanical property to identify differences based on gender, age, race, and facial location (*p* < 0.05). Repeatability was measured on every tenth subject, where biomechanical measurements were obtained at the same cheek location on the right (opposite) side of the face, and paired *t*-tests compared left and right-side measurements. To gain an understanding of properties representative of the overall subject population, two clustering analyses were performed.

To assess general viscoelastic behaviors, population clusters were derived from the properties of stiffness and creep. To assess the movement and the “feel” of facial skin, a second clustering analysis was applied to the properties of stiffness, elastic deformation, and absorbed energy. The four properties represented resistance to applied vacuum (stiffness), delayed deformation and damping (creep), tissue stretchability under continuous loading (elastic deformation), and texture (energy). For both clustering analyses, all age, sex, race, and facial location measurements were pooled for each biomechanical property. Ward’s minimum variance method was employed as clustering analysis to identify cluster values representative of the subject population [21].

### 2.2. Prosthetic Elastomer Measurements

Elastomers chosen for this study included those cited as the most used by clinicians in a published survey [22] and those with published tensile and hardness properties before and after outdoor weathering [23]. All materials were silica-filled polydimethyl siloxanes (PDMS) with Durometer hardness ranging from 20 (A2000) to 70 (A225-70) (Table 1).

Each product was purchased from Factor II (Lakeside, AZ, USA), mixed according to manufacturers’ instructions, subjected to 5 × 10^−^^3^ torr vacuum, poured into circular molds with dimensions 34 mm diameter × 3 mm thickness (*n* = 5), and heat polymerized at the temperature recommended for each product (83 °C to 120 °C). The mold was cooled, disc removed, and tested after 72 h to permit post-curing. The sample size was based on a previous study, where a significant difference (*p* < 0.05) of 3 units in Durometer hardness could be detected with 80% statistical power [23].

The same hand-held vacuum-generating instrument for measuring facial skin was used to measure viscoelastic properties of the elastomers. Each disc was centered on top of the glass chamber with a 20 g weight to provide adequate seal. The testing regimen applied to the disc was the same as that described for facial skin, and stress-deformation and deformation-time data were recorded. The same six biomechanical properties were determined.

For comparisons of elastomers with facial skin, the facial skin dataset was subdivided by facial location and biomechanical property measurements combined with those obtained for the elastomers. This permitted a direct comparison of a facial location with each elastomer for a given biomechanical property. Descriptive statistics, one-factor general linear model (GLM) and Tukey–Kramer post hoc tests determined significant differences among skin and elastomer group means for each biomechanical property (*p* < 0.05).

## 3. Results

### 3.1. Facial Skin Properties

#### 3.1.1. Results from Biomechanical Measurements

One study subject, a Hispanic female, age 30–49, signed an informed consent document, but did not participate in skin measurements. This reduced the subject population from 144 to 143 participants. Significant main effect contributions resulting from the linear mixed model caused by race, age, sex, and facial location on the six measured mechanical properties are presented in Table 2.

Facial location profoundly affected all properties (*p* < 0.0001), whereas race did not significantly contribute to any biomechanical property (0.13 ≤ *p* ≤ 0.96). Differences among age groups generated significant differences for percent laxity and creep (0.003 ≤ *p* ≤ 0.04), and the results are presented in Figure 3a,b.

The primary differences were associated with the 50–70 age group, where percent laxity was significantly lower than the 30–49 age group, and creep was higher than the 19–29 age group. Gender differences demonstrated significantly lower values for laxity, energy, and elastic deformation in females (0.02 ≤ *p* ≤ 0.049, Figure 3a,c,d), but none of the other measured properties were significantly affected (0.09 ≤ *p* ≤ 0.14). Effects rendered by the different facial locations on facial skin properties are summarized in Figure 3 and Table 3.

The body of ear skin was the most rigid and inflexible, as its values were highest for stiffness and lowest for creep, elastic deformation, percent laxity, and energy absorption. Cheek skin was nearly opposite in properties, as its stiffness was 1.8 times lower, whereas creep, elastic deformation, percent laxity, and energy were 1.6, 3.0, 2.5, and 2.0 times higher than the body of ear, respectively. Nose skin was similar to body of ear with its high stiffness, low laxity, low elastic deformation, and low energy, but it demonstrated 1.5 times more creep. Values measured for chin and forehead skin were intermediate to the aforementioned locations for the properties of elastic deformation, creep, and laxity, but forehead skin was stiffer. Ear lobe skin was highest in creep. Percent elasticity was calculated as:percent elasticity = (recovered deformation/elastic deformation) × 100(1)
and served as a measure of immediate elastic recovery after force release. Percent elasticity was highest for chin, cheek, and back of ear; intermediate for forehead and nose; and lowest for ear lobe.

Results from paired *t*-tests demonstrated no significant differences between left and right-side repeatability measurements (0.148 ≤ *p* ≤ 0.424, not shown).

#### 3.1.2. Results from Clustering Analyses

Hierarchical clustering analyses were performed as means to group biomechanical properties within a subject population so that subjects within a cluster were closer to each other than subjects grouped into different clusters [24]. The cluster center, or centroid, is defined by the mean values of properties incorporated within the cluster. Two important aspects of skin biomechanics are its viscoelasticity, or simultaneous stiffness and damping behaviors under load, and its movability when touched or in motion during joint function. Here, stiffness and creep were chosen to define population clusters for viscoelasticity. Stiffness, elastic deformation, and energy absorption were chosen to identify population clusters for facial skin stretchability and texture or “feel.” Wards analysis identified three clusters present within the subject population for both sets of analyses. Results are shown in Table 4, and the breakdown for the number of facial locations comprising each cluster is shown in Table 5.

For the viscoelastic properties of combined stiffness and creep, the subject population fell into three distinct clusters (Table 4, upper table). Cluster 1 contained the largest number of observations and its mean values represented a centroid with intermediate stiffness and creep. The mean values were similar to those reported for forehead and chin (Figure 3b, Table 3), and these two locations supplied the highest number of observations for Cluster 1 (Table 5). With the fewest observations, Cluster 2 contained measurements that mostly originated from the back of ear (Table 5) with stiffness being highest and creep lowest of the three clusters. Observations forming Cluster 3 were opposite of those for Cluster 2, as its mean values defining the centroid were lowest for stiffness and highest for creep, with the combination best representing cheek, nose, and ear lobe (Table 5).

For skin stretchability and texture, as represented by the properties of stiffness, elastic deformation, and energy, three distinct population clusters also were identified. The centroid results are presented in the lower table of Table 4. Cluster 1 was comprised of tissues that were highly stiff, while also being low in elastic deformation and absorbed energy. Most of the observations were located at the back of ear (Table 5). Conversely, tissues that were the most easily deformed and stretchable formed Cluster 3, where stiffness was lowest, and elastic deformation and energy being highest. This most closely corresponded to mean values reported for cheek (Figure 3c,d, Table 3), although Table 5 shows that the cluster also included large numbers of forehead, chin and ear lobe observations. Mean values shown for Cluster 2 in (lower) Table 4 were intermediate to those for Clusters 1 and 3. Observations comprising this cluster were largely measured at the tip of the nose, although nearly one-half of measurements taken from the ear lobe and back of ear fell into this cluster (Table 5).

### 3.2. Comparisons of Facial Skin Properties with Prosthetic Elastomer Properties

#### 3.2.1. Results from Biomechanical Measurements

Table 6 presents comparisons of biomechanical properties between facial skin (n = 143) and prosthetic elastomers (*n* = 5) from the one-factor GLM/Tukey.

Of the biomechanical properties measured, percent laxity showed the most similarities between elastomers and facial skin. Except for A2000, all elastomers were in the same statistical grouping as the back of ear. A2006, A2186, and A588-1 were grouped with chin, ear lobe, and nose, and both A2000 and A2186 were grouped with forehead skin. Only A2000 was in the same statistical grouping as cheek skin. However, these comparisons are misleading, as percentage laxity is a parameter that is normalized to elastic deformation. Consequently, comparisons should be made only within skin locations or within elastomers. Additional explanation is presented in Section 4.2.

For remaining properties, the prosthetic elastomers were vastly different from facial skin, with the exception of the back of the ear, which was grouped with the elastomers in elastic deformation and energy. Otherwise, the prosthetic materials were 1.8 to 6.4 times stiffer, 1.8 to 11.7 times lower in elastic deformation, 2 to 4 times lower in absorbed energy, and 2.75 to 9 times lower in viscous creep. Except for A2000, when vacuum was released, the prosthetic materials immediately recovered 74% to 84% of their original dimension, whereas facial skin recovered 32% to 50%—which was 1.5 to 2.5 times lower. Except for back of ear, facial skin was found to be significantly more flexible, stretchable, and viscous in its mechanical response.

#### 3.2.2. Comparisons with Clustering Analyses Results

To better visualize the mechanical differences between facial skin and the prosthetic elastomers, two- and three-dimensional scatterplots were constructed. Viscoelastic properties of stiffness and creep were plotted for the three subject population clusters and group of elastomers, and are shown in Figure 4a. For stretchability/texture properties, a 3D scatterplot is presented in Figure 4b.

In Figure 4a, the elastomers are scattered close to the abscissa, demonstrating low creep, and high stiffness compared to the facial skin clusters. Cluster 2 observations (red circles) are closest to the elastomers, and contain predominantly back of ear measurements (Table 5). For stretchability/texture properties shown in Figure 4b, the prosthetic elastomers demonstrate remarkably lower elastic deformation and energy compared to facial skin, and are closest to Cluster 1 (green circles), which contains mostly back of ear measurements (Table 5).

## 4. Discussion

The overall goal of research in this area is to restore the craniofacial complex to its full form and function following injury or disease. This project focused on mechanical property assessment of facial skin, as it determines the feel, comfort, and life-like qualities of a tissue replacement, be it biological, biomimetic, or a synthetic material. The process of successfully engineering replacements requires an understanding of existing properties of healthy tissues, comparing their properties with those that are present in currently used tissue replacements, then invoking an engineering design process to improve results. For prosthetic materials, strategies that impose changes to composition, processing, and placement are expected to be considered during the design process.

Results from this study both accepted and rejected the first null hypothesis, namely, that facial skin properties were not affected by facial location, age, sex, and gender. The hypothesis was accepted for race, partly accepted for age and sex (specific properties were affected), and rejected for facial location. The second hypothesis, that there are no differences in biomechanical properties between facial skin and a group of selected prosthetic materials, was rejected.

### 4.1. Biomechanical Measurements of Facial Skin

A wealth of information has been published from biomechanical tests of skin. Numerous test methods have been employed, and are summarized in reviews by Kalra et al. and Pierard et al. [25,26]. In this study, viscoelastic properties were measured with a device based on the application of vacuum. The device chosen for this study (BTC-2000) measures mechanics of both epidermis and dermis. This was considered relevant, as skin biomechanics is largely regulated by the composition and thickness of the epidermis, dermis, and hypodermis, with dermis being the major contributor [27]. This differs from a device often used (Cutometer), which has a smaller port opening (usually 2 mm to 6 mm, versus 10 mm for BTC-2000), and measures only the mechanical response of the epidermis. Barel [7] reported that devices with larger port openings, and hence contact with greater skin surface area, produce larger numbers for all measured biomechanical properties. Smalls et al. [27] measured six biomechanical properties in the leg, calf, and thigh with both the Cutometer and BTC-2000. They found that the properties of elastic deformation, energy, and elastic recovery measured with the BTC-2000 positively correlated with similar parameters measured with the Cutometer. Other properties either lacked correlation or were not common to both devices. Consequently, direct comparisons of results between the two instruments are limited.

Past research has identified age, sex, and body location as contributors to skin biomechanics. Race/ethnicity has been studied to a limited extent, with both differences and no differences in skin biomechanics reported. In this study of facial skin, the overwhelming contributor to property differences was facial location (all locations *p* < 0.0001), whereas race/ethnicity rendered little effect on any measured property (0.13 ≤ *p* ≤ 0.96). Age and sex affected two to three selected properties, otherwise their contributions were non-significant.

#### 4.1.1. Facial Location

Previous research has reported mechanical properties of facial tissues at a limited number of locations and compared the results with skin located elsewhere on the human body [5,7,19,28,29,30,31]. However, none have characterized multiple viscoelastic properties at six key locations for facial injury, a feature of this study, and only studies by Bellamy and Waters [17] and Farah et al. [18] have compared facial skin results with tissue replacements using the same experimental protocol and equipment. Data presented in Figure 3 and Table 3 show the body of ear and cheek to display almost the opposite mechanical behaviors when subjected to applied suction. Skin covering the ear body is thin (1 mm) [6] and tightly bound to underlying hyaline cartilage, which permits little movement and imparts high stiffness with minimum elastic deformation, absorbed energy, and creep. Cheek skin, on the other hand, is thicker (1.5–2 mm), and is more easily stretched/compressed due to its attachment to an adipose layer, which is approximately 5 mm thick [3]. With its loose connective tissue and numerous blood vessels, the fat permits ready skin movement over underlying facial musculature and continues to deform under constant load, thereby imparting flexibility and high creep. These two extremes in mechanics are punctuated by the fact that most ear body and cheek measurements occupy different clusters in Table 5, and create an indelible need to engineer separate tissue replacements capable of mimicking these structures.

The remaining facial locations tested in this study show skin properties that are overlapping and intermediate to those for cheek and ear body (Figure 3, Table 3, Table 4 and Table 5). Chin and forehead skin are thick (1.8–2.5 mm), with underlying fat layers that are one-half that of cheek skin. These skin locations are attached to bone, which serves as an unmovable anchor, permitting deformation of the overlying skin. As a result, elastic and energy properties are more similar to cheek skin than ear body; however, creep is intermediate to the two skin locations. Ear lobe skin is attached to elastic cartilage, which has a substantial elastin network that imparts intermediate flexibility and the ability to continue deforming under a static load, which accounts for its high creep. The tip of nose measurements for elastic properties of stiffness, elastic deformation, and energy are closer to ear body than to cheek and reflect skin attachment to hyaline cartilage. Interestingly, mean creep values are closest to cheek, and inspection of the stiffness-creep table in Table 5 show that nose observations to be nearly equally divided between Cluster 1 (intermediate creep) and Cluster 3 (high creep). The underlying reason for this behavior is unclear.

Laxity characterizes the tightness/looseness of skin and reflects the degree to which skin is bound to underlying tissue, indicating tendency to sag. With the BTC-2000, it is measured as the percentage of elastic deformation that occurs with very low stress and is defined by a change in slope of the stress-deformation curve (Figure 2). Healthy, young skin demonstrates low laxity, and increases in laxity with age are accompanied with wrinkling and folding. For the different facial locations, results presented in Figure 3a show that structures with increased skin and adipose thickness (cheek, forehead, and chin) exhibit higher laxity than those that are bound to cartilage (nose, ear lobe and ear body).

Percentage elasticity reflects the amount of recovered deformation that is normalized to elastic deformation and converted to percentage. It is an indicator of the degree to which skin is able to immediately recover its dimension when the pressure is released. Normalizing the recovered deformation to elastic deformation is sometimes helpful in comparing tissues that differ in dimension, particularly with respect to thickness. The viscoelastic nature of facial skin is evident, as little more than 50% of the deformation is immediately recovered for any facial location. Chin, cheek, and ear body skin demonstrated the highest elastic recovery, whereas ear lobe skin, with its high elastin content, only recovered 27% of its original elastic deformation.

#### 4.1.2. Age

Biomechanical tests of aging skin often cannot distinguish between chronological aging and photoaging, where the two different aging processes occur simultaneously. With chronological aging, histological changes include a disappearance of subepidermal oxytalan fibers, larger cystic spaces in the elastin matrix, and clumping of dense microfibrillar areas [32]. The skin becomes more flaccid, which is reflected by an increase in laxity and often manifested as a more wrinkled appearance. Thus, elastic and total deformation increase, elastic recovery decreases, and viscous creep increases [7,29,33]. The tensile relaxation also changes the orientation of Langer’s lines, and the degree of anisotropy increases [34,35]. In photoaging, the elastic fiber network is additionally disrupted through an accumulation of granular elastotic material and accumulated glycosaminoglycans. Echography has shown these accumulations produce subepidermal bands that increase facial skin thickness from 0.1 mm to 0.5 mm [4]. As a result, when chronological aging is accompanied by photoaging (which is expected in highly solar exposed areas such as the face), the reported biomechanical changes include increased elastic modulus, decreased elastic recovery, and increased viscoelastic deformation [7,28,36]. Results for this subject population show agreement for creep as viscous deformation increased with age (Figure 3b). However, age did not significantly affect elastic deformation, stiffness, energy, or percent elasticity, which is in contradiction to the above discussion. This may be partly explained by the fact that this study performed tests on a stratified subject population, where observations are evenly balanced among population subgroups. Most prior research has focused on populations that are limited in number and/or do not include equal numbers of measurements for different genders, ages, races/ethnicities, and body locations. Additionally, differences in measurement equipment and protocols may contribute to these disparities. For percentage laxity, the 50–70 age group was significantly lower than the 19–29 age group, which is perplexing. No explanation can be offered for this occurrence.

#### 4.1.3. Sex

Results from published research suggest that sex contribution to skin biomechanics appears to be minimal, as tests of forearm, forehead, hand, and chest skin have demonstrated no significant differences between sexes for elastic modulus, elastic deformation, and elastic recovery [30,37,38]. In a comprehensive report of biomechanical tests conducted using the vacuum method, Barel [7] observed no gender differences for viscoelasticity ratio (creep divided by elastic deformation) and overall elasticity (similar to percentage elasticity) at four different anatomical locations. These results support those previously reported by Cua [29]. Compared to males, female skin extensibility has been reported as higher and the modulus of elasticity is lower, but the reverse has been reported as well [26,39,40]. In this study, male skin was significantly higher in laxity, elastic deformation, and absorbed energy (Figure 3a–c). These differences may be attributed to morphological differences, where the dermis is thicker in males and provides increased volume for displacement under vacuum. Stiffness and creep were not significantly different between males and females (Table 2), which is consistent with measurements reported for other body locations [19,29,37].

#### 4.1.4. Race

Few studies have compared biomechanical properties of skin based on race. In a small male population, a 15 kN m rotating torque was applied to forearms of Black, Hispanic, and White subjects. Differences in elastic recovery and extensibility were noted between the dorsal and ventral surfaces for Hispanic and White subjects, but not Black subjects. Melanin protection on dorsal surfaces of Black subjects was attributed to these differences. Black skin demonstrated 26% less elastic recovery compared to other groups [41]. In another study of skin elasticity, the opposite results were found, where no differences in elastic recovery were noted between Black and White subjects on the legs, but skin recovery was 1.5 times higher for Black subjects on the face [42]. Discrepancies between the studies were ascribed to differences in age groups studied, but overall, the results from the two studies have been deemed inconclusive [43]. In this study, none of the six biomechanical properties were significantly affected at any facial location based on differences in race/ethnicity (Table 2). Compared to other studies, this study enlisted a larger subject population that was equally stratified for gender, age, and race/ethnicity. Along with a different measurement method, this likely accounts for differences in reported results.

### 4.2. Facial Prosthetic Elastomers

Comparisons of prosthetic elastomers with facial skin properties presented in Table 6 show the materials to be categorically highly rigid with low deformability, stretchability, and viscous behavior compared to facial skin. Only the ear body, with its thin skin layer attached to hyaline cartilage, behaved similarly as the properties of elastic deformation, absorbed energy, and percent elasticity overlapped with several prosthetic materials. The disparities measured for creep were particularly remarkable as the prosthetic materials were nearly an order of magnitude lower. Similarly, the elastomers exhibited a high degree of elastic recovery following stress removal, between 75% and 85% for all except A2000, whereas facial skin showed only 28% to 49%. This speaks to the limitations that are provided by a filled and crosslinked elastic polymer network. Skin has water and GAGs that can flow through collagen and elastin networks under stress, and time-dependent diffusion is required to restore the original tissue dimension following stress removal. The flow-like behavior afforded in the elastomer network is limited to the hysteresis created during unloading, with unrecovered strain remaining when the load falls to zero. The comparisons of prosthetic materials with population clusters shown in Figure 4 visually depict the seemingly one-dimensional nature of the prosthetic elastomer system which produces materials that vary widely in stiffness, but its ability to produce a range of elastic deformations, creep, and absorbed energies representative of facial skin is minimal.

It is interesting that the prosthetic materials overlapped with facial skin in percent laxity, particularly since material stiffness was up to 6.4 times higher and elastic deformation 11.7 times lower. This is attributable to the method by which laxity is measured and percent laxity calculated. As stated, laxity is measured in millimeters at the point where the stress-deformation curve changes slope, and percent laxity is calculated by dividing this deformation by the amount of elastic deformation, then multiplying it by 100. Since the value is normalized to the amount of elastic deformation, similar values can be obtained for stiff surfaces exhibiting low elastic deformation, and flexible surfaces with high elastic deformation. Therefore, this property is useful only for comparing similar systems, such as skin to itself and elastomers to themselves. Comparing dissimilar systems (i.e., skin versus prosthetic materials) can lead to erroneous conclusions.

### 4.3. Engineering Design Considerations

An inspection of facial skin biomechanical properties at various locations and within population clusters suggests that, for the facial locations measured in this study, three types of tissue replacements are required. As discussed, the body of ear and cheek properties are distinctly different from other facial locations, and therefore require separate considerations. Consequently, each location requires the engineering of a skin replacement with its own properties, particularly with respect to stiffness and creep. Property values approaching those listed in Figure 3 and Table 3 for each site would serve as guidelines for the design. Although nose, chin, ear lobe, and forehead skins exhibit their own unique biomechanical behaviors, a considerable overlap in properties is present. Therefore, it seems reasonable to choose a skin replacement that targets properties listed for Cluster 1 (upper table) and Cluster 2 (lower table) of Table 4 for best satisfying the mechanical behaviors of skin present at these facial locations.

Compared to their predecessors, filled PDMS materials have been used as facial prosthetic materials for nearly five decades because of their flexibility, ease of fabrication, biocompatibility, and relative color stability. Based on results presented here, their biomechanical properties fall short of matching those measured for skin at most facial locations. Part of this is owed to the fact that the present processing methods for facial prostheses are able to accommodate only a single material to serve as the entire prosthesis. Facial structures are multi-layered and exhibit a gradation of properties from outer to inner structures. The successful engineering of tissue replacements, be they tissue-engineered, biomimetic, or prosthetic, require an approach that permits the layering of components to produce a composite structure that more closely duplicates the biomechanical behavior of tissue at a given facial location. This very strategy has been studied by Bellamy and Waters [17]. Using uniaxial linear extensometry, facial skin was stretched in undisclosed locations in 15 volunteers, and the results were applied to construct a three-layered composite material. The middle layer contained varying ratios of unreactive silicone fluids and polydimethyl siloxane, and certain formulations produced similar force-decay results as compared to those measured for facial skin. However, similarities were not seen in hysteresis behavior, and the authors concluded continued research was needed. These results demonstrate that a multilayered design holds promise for future facial prosthetic material development.

For prosthetic materials, changes are needed in both material processing and composition. Additive manufacturing is a logical strategy for producing layered structures, and it has been used to produce components for facial reconstruction [44,45,46]. However, these are mostly single composition materials that focus solely on hard or soft tissue replacement, and often without direct comparison to known properties of healthy tissues. Furthermore, advancements in 3D printer development that permit the layering of rigid and flexible polymers that coincide with future material development are needed [47]. For filled PDMS, continued strategies to lower stiffness and increase deformability that include copolymerizing comonomers that widely differ in molecular weight, altering the monomer: crosslinker ratios, exploring combinations of nano- and submicron–sized fillers at various loading levels, and developing filler coatings that promote superior filler dispersion within polymer are needed. Another approach is to combine cellulose fibers with polyisoprene to mimic collagen and elastin networks in skin, which has produced elastomers shown to exhibit nonlinear stress–strain behaviors similar to those reported for ex vivo tests of human skin [48]. All strategies must include considerations that accommodate requirements for fluid rheology during printing, polymerization via ultraviolet and laser energy sources [49,50], and biocompatibility [51].

### 4.4. Study Limitations

Results from this study are limited by patient variability, testing protocols, equipment, and material batches. Inclusion criteria included healthy appearing facial skin, but other factors, such as patient health and habits (e.g., tobacco usage), were not measured nor controlled, which may impact skin mechanics. The vacuum testing protocol was designed to measure creep over ten seconds as a means to obtain sustained viscous behavior, but also was longer than in previous works where shorter time periods were used and a greater elastic response was produced. This limits the ability to compare this study’s results with others regarding the relative contributions from elastic and viscous components of the viscoelastic response. As stated earlier, the BTC-2000 equipment permits measurement of dermal contributions to skin mechanics, but also limits comparisons with studies enlisting other measurement modes. For prosthetic elastomer materials, batch differences are not expected to yield pronounced changes in properties, but these differences may contribute to the disparate results reported among different laboratories.

## 5. Conclusions

Six biomechanical properties of skin were measured at six locations on the faces of 143 human subjects, and the results were compared to those measured for eight prosthetic elastomers currently available for clinical usage. Generally, the elastomers demonstrated a similarity to body of ear skin for the properties of elastic deformation, absorbed energy, and percentage elasticity. Otherwise, the prosthetic materials were significantly different from skin measured at other facial locations, with the most disparate differences noted for stiffness and creep, two key viscoelastic properties. To engineer facial skin replacements, separate considerations are needed for the ear body and cheek, whereas a similar set of biomechanical parameters can be chosen to satisfy those measured for chin, nose, forehead, and ear lobe.

## Figures and Tables

**Figure 1 materials-16-02023-f001:**
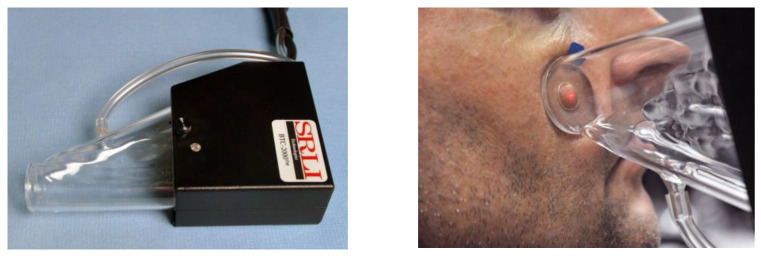
**Left**: Glass chamber used to create vacuum and measure displacement with skin or elastomer. **Right***:* Vacuum applied to facial skin, with displacement measured by movement of laser target, which is seen as a red dot in the center of displaced skin.

**Figure 2 materials-16-02023-f002:**
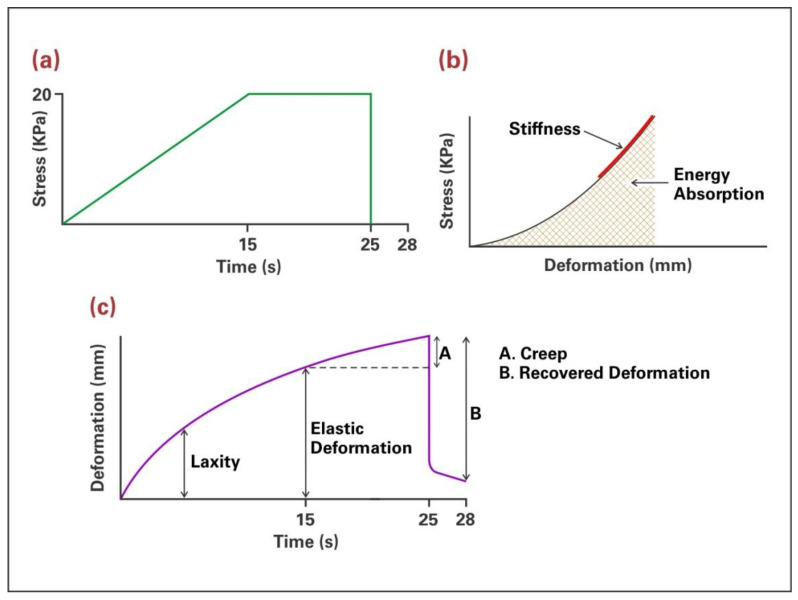
(**a**) Stress-time plot showing loading regimen of applied vacuum. (**b**) Stress-deformation plot showing stiffness measured as the slope of the straight-line portion and energy measured as the area under the plot. (**c**) Deformation-time plot with laxity measured as initial deformation produced by low stress, elastic deformation being the deformation accumulated at the end of vacuum application, creep (A) representing viscous deformation occurring while vacuum is held constant and recovered deformation (B) as the amount of deformation recovered over 3 s following vacuum release. Percent laxity is calculated as laxity divided by elastic deformation times 100. Percent elasticity is calculated as recovered deformation divided by elastic deformation times 100.

**Figure 3 materials-16-02023-f003:**
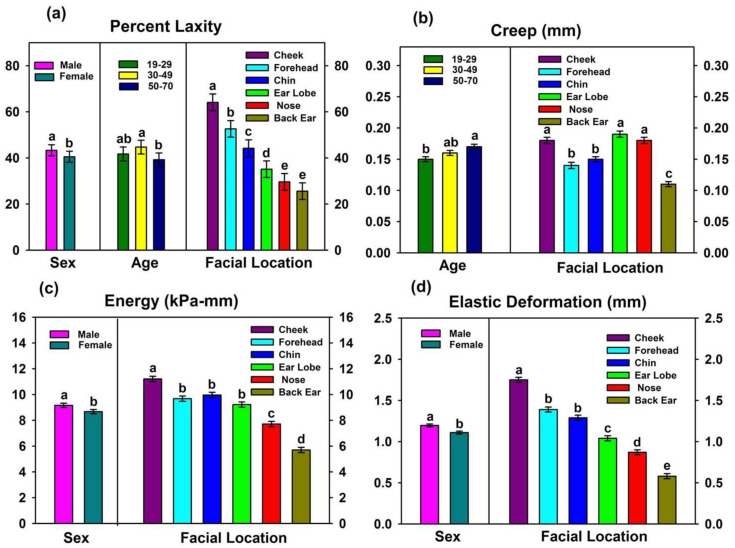
Main effects of facial location, gender, and age on (**a**) laxity, (**b**) creep, (**c**) energy absorption, and (**d**) elastic deformation. Groups denoted with the same lower-case letters are not significantly different from each other (*p* ≥ 0.05). Comparisons are only within each main effect. Error bars represent standard error of means.

**Figure 4 materials-16-02023-f004:**
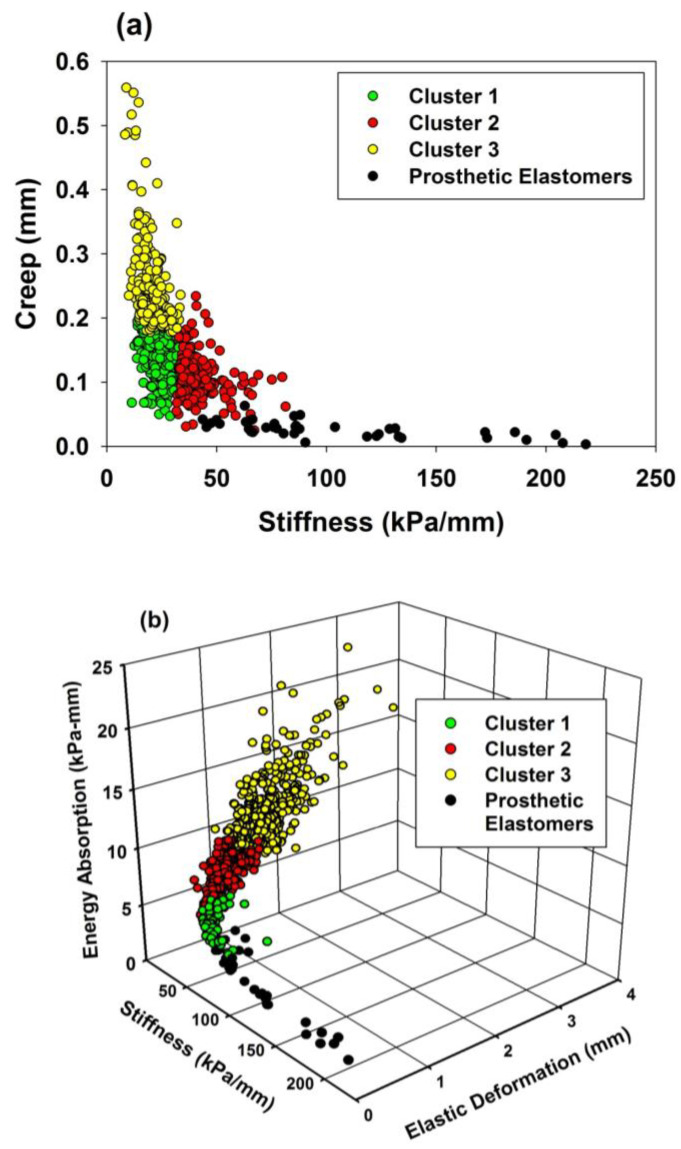
(**a**) Two-dimensional scatterplot of stiffness and creep for facial skin clusters and prosthetic elastomers. (**b**) Three-dimensional scatterplot of stiffness, elastic deformation, and energy for facial skin clusters and prosthetic elastomers.

**Table 1 materials-16-02023-t001:** Prosthetic Elastomers Tested in This Study.

Product	Batch Number
A2000	A54615, B54615
A2006	S120511
A2186	F62846
A588-1	S01112
A223-30	S 14U132L06
A225-50	F 51110110
A225-70	S 14U109L03
RTV-40	S41112136

**Table 2 materials-16-02023-t002:** Main effect contributions by facial location, race, age, and sex to measured mechanical properties. *p* values from linear mixed model (* *p* < 0.05).

Mechanical Property	Facial Location	Race	Age	Sex
Laxity	<0.0001 *	0.60	0.008 *	0.049 *
Stiffness	<0.0001 *	0.43	0.72	0.11
Elastic Deformation	<0.0001 *	0.85	0.68	0.02 *
Creep	<0.0001 *	0.96	0.04 *	0.14
Energy	<0.0001 *	0.67	0.84	0.049 *
Recovered deformation	<0.0001 *	0.13	0.175	0.09

**Table 3 materials-16-02023-t003:** Comparisons of stiffness and percent elasticity measured at different facial locations (mean *±* se) *.

Facial Location	Stiffness (kPa/mm)	Percent Elasticity
Back of Ear	39.8 ± 0.72 ^A^	47.1 ± 1.04 ^B^
Nose	27.3 ± 0.73 ^B^	31.7 ± 1.03 ^D^
Forehead	26.2 ± 0.71 ^B^	38.2 ± 1.03 ^C^
Ear Lobe	25.4 ± 0.72 ^BC^	28.7 ± 1.02 ^D^
Chin	23.7 ± 0.72 ^CD^	50.6 ± 1.03 ^A^
Cheek	22.4 ± 0.71 ^D^	48.0 ± 1.04 ^AB^

* Means with the same superscript letter are not significantly different (*p* ≥ 0.05). Vertical comparisons only.

**Table 4 materials-16-02023-t004:** Subject population clusters representing viscoelasticity and stretchability of facial skin with facial locations pooled.

Viscoelastic Properties of Stiffness and Creep					
Cluster	Observations	Property	Mean ± SE	Minimum	Maximum
1	442	Stiffness (kPa/mm) Creep (mm)	24.8 ± 0.21 0.14 ± 0.001	11.3 0.05	33.5 0.20
2	184	Stiffness (kPa/mm) Creep (mm)	42.5 ± 0.79 0.11 ± 0.003	31.7 0.02	81.4 0.19
3	232	Stiffness (kPa/mm) Creep (mm)	20.7 ± 0.35 0.25 ± 0.005	8.3 0.18	33.6 0.56
**Stretchability/Texture Properties of Stiffness, Elastic Deformation, and Energy**					
**Cluster**	**Observations**	**Property**	**Mean ± SE**	**Minimum**	**Maximum**
1	113	Stiffness (kPa/mm)	46.7 ± 0.91	36.7	81.4
		Elastic Defm (mm)	0.5 ± 0.01	0.2	0.8
		Energy (kPa·mm)	4.8 ± 0.06	3.5	6.0
2	340	Stiffness (kPa/mm)	28.7 ± 0.24	19.6	39.9
		Elastic Defm (mm)	0.8 ± 0.01	0.4	1.4
		Energy (kPa·mm)	7.3 ± 0.06	4.8	9.8
3	405	Stiffness (kPa/mm)	20.9 ± 0.23	8.3	38.0
		Elastic Defm (mm)	1.6 ± 0.02	0.9	3.6
		Energy (kPa·mm)	11.1 ± 0.11	8.0	22.7

**Table 5 materials-16-02023-t005:** Number of observations for each facial location per cluster for viscoelastic and stretchability/texture properties.

	Stiffness, Creep			Stiffness, Elastic Deformation, and Energy		
Facial Location	Cluster 1	Cluster 2	Cluster 3	Cluster 1	Cluster 2	Cluster 3
Cheek	87	2	54	0	10	133
Forehead	136	17	20	6	40	97
Chin	102	14	27	3	51	89
Ear Lobe	52	28	63	22	60	61
Nose	56	24	63	11	110	22
Back of Ear	40	100	6	71	69	3

**Table 6 materials-16-02023-t006:** Biomechanical properties comparisons among facial skin and prosthetic elastomers *.

Skin/Elastomer	Laxity ^1^ (Percent)	Stiffness (kPa/mm)	Elastic Deformation (mm)	Energy (kPa-mm)	Creep (mm)	Percent Elasticity
Cheek	64.1 ± 1.49 ^A^	22.4 ± 0.71 ^F^	1.75 ± 0.031 ^A^	11.2 ± 0.21 ^A^	0.18 ± 0.005 ^A^	48.0 ± 0.94 ^B^
Forehead	52.6 ± 1.48 ^B^	26.2 ± 0.71 ^E^	1.39 ± 0.030 ^B^	9.7 ± 0.22 ^B^	0.14 ± 0.005 ^B^	38.2 ± 1.19 ^C^
Chin	44.2 ± 1.49 ^C^	23.7 ± 0.72 ^E^	1.29 ± 0.031 ^B^	10.0 ± 0.21 ^B^	0.15 ± 0.004 ^B^	50.6 ± 1.33 ^B^
Ear Lobe	35.1 ± 1.47 ^D^	25.4 ± 0.72 ^E^	1.04 ± 0.031 ^C^	9.2 ± 0.19 ^B^	0.19 ± 0.005 ^A^	28.7 ± 0.82 ^D^
Nose	29.6 ± 1.49 ^E^	27.3 ± 0.73 ^E^	0.87 ± 0.030 ^D^	7.7 ± 0.23 ^C^	0.18 ± 0.005 ^A^	31.7 ± 1.03 ^D^
Back of Ear	25.6 ± 1.48 ^E^	39.8 ± 0.72 ^D^	0.58 ± 0.031 ^E^	5.7 ± 0.21 ^D^	0.11 ± 0.004 ^C^	47.1 ± 1.12 ^B^
A2000	58.1 ± 7.32 ^ABCD^	73.6 ± 3.74 ^C^	0.47 ± 0.047 ^E^	3.9 ± 0.19 ^D^	0.03 ± 0.002 ^D^	46.2 ± 5.63 ^B^
A2006	22.1 ± 9.60 ^CDE^	60.0 ± 8.13 ^C^	0.34 ± 0.043 ^E^	4.0 ± 0.11 ^D^	0.04 ± 0.009 ^D^	82.9 ± 4.70 ^A^
A2186	36.1 ± 3.43 ^BCDE^	80.9 ± 4.22 ^C^	0.23 ± 0.002 ^E^	3.2 ± 0.14 ^D^	0.02 ± 0.002 ^D^	84.5 ± 3.03 ^A^
A588-1	23.4 ± 9.38 ^CDE^	64.1 ± 8.43 ^C^	0.32 ± 0.035 ^E^	3.7 ± 0.20 ^D^	0.04 ± 0.002 ^D^	81.0 ± 0.47 ^A^
A223-30	18.0 ± 9.00 ^E^	78.9 ± 7.34 ^C^	0.23 ± 0.019 ^E^	3.0 ± 0.10 ^D^	0.04 ± 0.003 ^D^	74.5 ± 1.42 ^A^
A225-50	14.0 ± 8.63 ^E^	151.5 ± 12.87 ^B^	0.12 ± 0.014 ^E^	2.3 ± 0.78 ^D^	0.02 ± 0.004 ^D^	79.0 ± 3.57 ^A^
A225-70	22.1 ± 7.08 ^E^	218.0 ± 30.8 ^A^	0.08 ± 0.006 ^E^	3.0 ± 0.23 ^D^	0.01 ± 0.003 ^D^	75.9 ± 5.90 ^A^
RTV-40	27.2 ± 4.87 ^E^	144.6 ± 14.23 ^B^	0.15 ± 0.008 ^E^	2.8 ± 0.05 ^D^	0.02 ± 0.001 ^D^	77.7 ± 4.13 ^A^

* Groups denoted with the same uppercase letter are not significantly different from each other (*p*
> 0.05). Comparisons are vertical only, and include both facial skin and elastomers. ^1^ See Section 4.2 regarding comparisons of skin and materials for this property.

## Data Availability

Non-human data are available from the corresponding author upon request.

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
