# Peer review of "Viscoelastic Properties of Human Facial Skin and Comparisons with Facial Prosthetic Elastomers"

_materials, 2023, doi:10.3390/ma16052023_

Round 1

Reviewer 1 Report

The paper aims to measure viscoelastic skin properties at six different locations on the faces of 143 human subject to provide baseline information for designing future replacements for missing facial tissues. The same properties were also measured for a group of polymeric prosthetics materials currently available for clinical usage, to compare if such prosthetics are actually able to duplicate the appearance, biology and function of the native tissue.

The entire process of data acquisition and data processing is well described and robust. The number of subjects involved in the data collection phase is adequate.

The results are presented clearly and comprehensively, effectively supporting discussion and conclusions.

However, some minor issues should be considered by the authors to improve the paper:

-          it would be interesting to compare the results obtained by the authors with the existing data published in the articles cited in the text (lines 63 – 65).

-          The first paragraphs of sections 4.1, 4.1.1,4.1.4 and 4.2 should be moved and integrated into the Introduction and Materials and Methods, to justify the authors' choices regarding both the data collection protocol and the choice of facial prosthetic elastomers.

-          In line 330, authors should be most specific referring to “past research” by providing more specific information

Author Response

The authors thank the reviewer for taking time from a busy schedule to carefully consider the manuscript and provide constructive commentary that should strengthen the revised document.  Responses to specific comments are provided below.

-          it would be interesting to compare the results obtained by the authors with the existing data published in the articles cited in the text (lines 63 – 65).

We looked at doing this, but the studies were vastly different. The cadaveric study tested dumbbells in tension (ref 16 in revised version), Bellamy’s work employed a custom-made linear extensometry set up (ref 17) and early work by Farah (ref 18 ) used an indentation technique. Other studies used the cutometer at different facial locations or measured skin mechanics with custom-made devices, and evaluated only women, only one race, limited age range, etc. Facial location, loading rates, testing modes, subject population – nothing matches up among their studies, or with ours. Agreed, this would have been a good opportunity to compare consistencies/inconsistencies in results if enough similarities in study design were present.

-          The first paragraphs of sections 4.1, 4.1.1,4.1.4 and 4.2 should be moved and integrated into the Introduction and Materials and Methods, to justify the authors' choices regarding both the data collection protocol and the choice of facial prosthetic elastomers.

Thank you – these are very helpful suggestions for improving the document’s organization. 1st paragraphs of Sections 4.1 and 4.1.1 have been inserted into the  introduction, except for the last two sentences of 4.1.1, which were retained in that section to provide clarity. The 1st paragraph of Section 4.2 has been moved to the 1st paragraph of section 2.2 in Materials and Methods. Section 4.1.4 was not moved since it did not flow with existing text and to do so would remove information relevant to the discussion section.

 -          In line 330, authors should be most specific referring to “past research” by providing more specific information

Section 4.1 is introductory and provides a lead-in to the sections that follow. Line 330 in the original manuscript is written as a general statement, with the contributions of facial location, age, sex and race being specifically addressed in sections that immediately follow, 4.1.1 through 4.1.4.

Reviewer 2 Report

Well structured and executed scientific article. Just a few criticisms listed below:

-LINE 28 define, indicating them, the tests used in the formulation of the study

-check that all keywords are pubmed mesh terms

How was the sample size calculated?

- some considerations should be performed on the biocompatibility of these devices. In this regard, I suggest to insert in the reference section the following scientific work that could be of help to the reader:

Pagano S, Lombardo G, Caponi S, et al. Bio-mechanical characterization of a CAD/CAM PMMA resin for digital removable prostheses. Dent Mater. 2021;37(3):e118-e130. doi:10.1016/j.dental.2020.11.003

- a section on the limitations of the study is missing

Author Response

The authors thank the reviewer for taking time from a busy schedule to carefully consider the manuscript and provide constructive commentary that should strengthen the revised document.  Responses to specific comments are provided below.

-LINE 28 define, indicating them, the tests used in the formulation of the study

The six properties are now stated in the abstract.

-check that all keywords are pubmed mesh terms

Yes, all are pubmed mesh terms. This was checked prior to manuscript submission.

How was the sample size calculated?

Thank you for catching this omission. For facial skin, the sample size was calculated from power analyses of preliminary trial data to detect a significant difference (p<0.05) of 3 kPa/mm stiffness at different facial locations with 80% statistical power. For prosthetic elastomers, the sample size was based on a previous study where, for a similar group of materials, a significant difference (p<0.05) of 3 units in Durometer hardness could be detected with 80% statistical power (reference 23 in revised version). This information has been added to the manuscript text. See the second sentence in section 2.1, and the last sentence in paragraph 2 of section 2.2., in the revised manuscript.

- some considerations should be performed on the biocompatibility of these devices. In this regard, I suggest to insert in the reference section the following scientific work that could be of help to the reader:

Pagano S, Lombardo G, Caponi S, et al. Bio-mechanical characterization of a CAD/CAM PMMA resin for digital removable prostheses. Dent Mater. 2021;37(3):e118-e130. doi:10.1016/j.dental.2020.11.003

The reference has been added. Please see citation #51.

- a section on the limitations of the study is missing

Again, thank you for catching this omission. Section 4.4 on Study Limitations has been added.

Reviewer 3 Report

Dear Author, 

Please view the attachment 

Regards

Author Response

The authors thank the reviewer for taking time from a busy schedule to carefully consider the manuscript and provide constructive commentary that should strengthen the revised document.  Responses to specific comments are provided below.

  1. It is mentioned that - these prostheses are based on filled poly(dimethyl siloxanes) or polyurethanes that produce prostheses… kindly revise

The authors are uncertain as to whether the reviewer wishes additional materials to be included, or the wording to be changed to yield a more accurate statement. The word “produce” has been changed to “render” in the above sentence, but additional changes are welcomed if the reviewer can be more specific.

  1. 144 human subjects were enrolled and stratified to produce equal numbers of subjects represented by gender (male, female), age (19-29, 30-49, 50-70) and race/ethnicity (Asian, African-American, Hispanic, White) (n=6). Sample distribution is not clear. Kindly mention whether it is in respective of the different locations or the different tests evaluated

Another reviewer requested information regarding sample size determination, and this was based on preliminary data obtained from different facial locations. A statement to this effect has been added as the second statement appearing in section 2.1 in the revised manuscript.

When subdividing 144 subjects into each subgroup for gender (2), age (3) and race/ethnicity (4), each cell contains 6 subjects.

  1. It is mentioned that- and only two studies have compared facial skin results with tissue replacements using the same experimental protocol and equipment [10,11]… kindly cite the authors

Agreed! Yes, the work of Farah et al and Bellamy & Waters are landmark studies, and these scientists should be fully recognized. Their names now appear within the text. See the second sentence of section 4.1.1, and references [17], [18] in the revised manuscript.

  1. It is mentioned that- Sex contribution to skin biomechanics appears to be minimal, as tests of forearm, forehead, hand and chest skin have demonstrated no significant differences between sexes for elastic modulus, elastic deformation and elastic recovery… kindly revise the narration as demarcation from present study is required

Thank you for pointing this out. Text has been reworded to discern published data from those reported in this work. This appears in the first sentence of section 4.1.3.